

# Survival between synchronous and non-synchronous multiple primary cutaneous melanomas—a SEER database analysis

Jie Xiong[1,2], Yanlin Su[3], Zhitong Bing[4] and Bihai Zhao[1]

[1] Department of Mathematics and Computer Science, Changsha University, Changsha, Hunan, China
[2] Department of Epidemiology and Health Statistics, Central South University, Changsha, Hunan, China
[3] Department of Gynaecology and Obstetrics, Changsha Central Hospital, Changsha, Hunan, China
[4] Evidence Based Medicine Center, Lanzhou University, Lanzhou, Gansu, China

Corresponding authors
Bihai Zhao, bihaizhao@163.com
Jie Xiong, xiongjie86@126.com

## ABSTRACT

**Background:** There is no criterion to distinguish synchronous and non-synchronous multiple primary cutaneous melanomas (MPMs). This study aimed to distinguish synchronous and non-synchronous MPMs and compare the survivals of them using the Surveillance, Epidemiology, and End Results database.

**Methods:** Synchronous and non-synchronous MPMs were distinguished by fitting the double log transformed distribution of the time interval between the first and second primary cutaneous melanomas (TIFtS) through a piecewise linear regression. The overall and melanoma-specific survivals were compared by the Kaplan–Meier method and Cox proportional hazard model through modeling the occurrence of synchronous MPMs as a time-dependent variable.

**Results:** The distribution of TIFtS was composed by three power-law distributions. According to its first inflection point, synchronous MPMs were defined as tumors that occurred within 2 months. The Kaplain–Meier plot revealed a significant inferior survival for synchronous MPMs than non-synchronous MPMs ($P < 0.0001$), and the occurrence of synchronous MPM was a risk factor for overall survival of cutaneous melanoma (CM) (hazard ratio: 2.213; (95% CI [2.087–2.346]); $P < 0.0001$).

**Conclusions:** This study provided data analysis evidences for using 2 months to distinguish synchronous MPMs and non-synchronous MPMs. Furthermore, the occurrence of synchronous MPM was a risk factor for prognosis of patients with CM.

## INTRODUCTION

Cutaneous melanoma (CM) is the most lethal type of skin cancer. The incidence, mortality, and disease burden of CM have been increasing annually (*Ali, Yousaf & Larkin, 2013*; *GBD 2015 Mortality & Causes of Death Collaborators, 2016*). In 2019, it is estimated that there will be 96,480 new cases of CM and an estimated 7,230 people will die from

the disease in the United States (*National Cancer Institute, 2019*). Although most CMs are initially diagnosed as localized and the 5 year survival rate is high (*Bradford et al., 2010*), one-third of all CM patients will experience disease recurrence and about 10–40% of patients diagnosed with localized lesions die from CM eventually (*Soong et al., 1998*; *Hanniford et al., 2015*). Therefore, it is particularly important to identify and monitor patients who have already had CM in order to detect subsequent CMs as early as possible (*Ferreres, Moreno & Marcoval, 2009*). In the clinical research of subsequent CMs, survival comparison between patients with multiple primary CM (MPM) and single primary CM (SPM) is an old question (*Hwa et al., 2012*). Many studies have been carried out to address this problem but the results are controversial (*Hwa et al., 2012*; *Utjés et al., 2017*; *Savoia et al., 2012*). Recently, *Grossman et al. (2018)* revealed the potential reasons for these controversies by analyzing the Surveillance, Epidemiology, and End Results (SEER) data using a single matching method.

However, MPM includes both synchronous MPM and non-synchronous MPM. Synchronous MPM is a subgroup of MPM, in which two or more primary tumors are detected simultaneously, and non-synchronous MPM is initially diagnosed as SPM until the second subsequent primary CM is detected in the follow-up. In previous studies (*Hwa et al., 2012*; *Utjés et al., 2017*; *Savoia et al., 2012*; *Grossman et al., 2018*), synchronous MPMs were either mixed with non-synchronous MPMs or discarded. Thus, the survival of synchronous MPM is not yet known. Furthermore, more importantly, how to distinguish synchronous MPM and non-synchronous MPM from the time interval between the first and the second primary CMs (TIFtS) is still unclear and arbitrary. *Grossman et al. (2018)*, *Pomerantz, Huang & Weinstock (2015)*, and *Moseley et al. (1979)*, adopted 1 year, 2 months, and 3 months to exclude synchronous MPMs, respectively. On the one hand, there is obviously no reason to believe that two primary CMs are synchronous with each other if the second primary CM occurred 1 year after the first one. Thus, 1 year is long enough to exclude synchronous MPMs; however, a longer TIFtS may also exclude more non-synchronous MPMs. On the other hand, are 2 months or 3 months long enough to exclude synchronous MPMs? If not, this may include some synchronous MPMs.

Herein, we explored the distribution of TIFtS using the SEER database to distinguish synchronous MPM and non-synchronous MPM. Based on this distinguishment, survivals between synchronous MPM and non-synchronous MPM were compared.

## MATERIALS AND METHODS

Both microscopically confirmed in situ and malignant CMs were retrieved from the SEER 18 program (1975–2016) (*National Cancer Institute, 2019*). The patients were followed up until December 2016. White patients with known age and at least two primary CMs were included in this study, while patients without both the first and second primary CMs were excluded. Furthermore, due to the high 5 year survival rate of CM and to ensure patients have enough time to develop subsequent primary CMs, patients with at least 5 years follow-up were included. Thus, patients first diagnosed from 1975 to 2011 and their subsequent primary CMs occurred in 2012–2016 were included. Patients first

diagnosed from 2012 to 2016 were excluded. Finally, Patients with unknown survival time were excluded from this study. Our study was exempt from institutional review board oversight, because the SEER 18 database is accessible to the public and the patients in the database are de-identified.

We calculated the TIFtS for each patient and the distribution of TIFtS was double log transformed. A piecewise linear regression, which is implemented by the "segemented" R package (*Muggeo, 2008*), was used to fit the double log transformed distribution. The confidential intervals of the cut points were also estimated by the "segemented" R package. Because synchronous MPMs should be near each other, thus, the first regressed line was defined as synchronous MPMs and the first cut point was defined as the optimal time to distinguish synchronous and non-synchronous MPMs. Furthermore, as occurrences of subsequent primary CMs were time dependent, we modeled the occurrence of subsequent primary CM as a time dependent variable and pre-processed the survival data into a start–stop format. The validity of this approach can be derived from the counting process theory of partial likelihoods (*Dirk, 2016*). Finally, Overall survival and CM-specific survival were compared.

All analyses were conducted by R software (version 3.4.4) (*Ihaka & Gentleman, 1996*). Survivals were compared by Kaplain–Meier method and Cox proportional hazard models. $P$ value $< 0.05$ was considered to reject the null hypothesis.

## RESULTS

In the SEER 18 database, 128,746 CM patients diagnosed from 1975 to 2011 have developed MPMs including 187,054 primary CMs, in which 19,924 subsequent primary CMs were detected from 2012 to 2016. Furthermore, 96,910 patients didn't have both the first and the second primary CMs (112,481 tumors). After filtering, 31,836 MPM patients were firstly included in this study to investigate the distribution of TIFtS. A kernel density estimation analysis showed that the distribution of TIFtS looks like comprised by three power-law distribution (Fig. 1A). Thus, we transformed the distribution of TIFtS into double log coordinates, and a piecewise linear regression was adopted to fit the double log transformed distribution. The result showed that there were three patterns that represented by three regression lines, respectively (model R square: 0.956, Fig. 1B). For the first regression line, the inflection point was at 2 months (95% CI [2.53–3.72] months), and we choose this time point to distinguish synchronous and non-synchronous MPMs. Interestingly, this agrees with the experience of *Pomerantz, Huang & Weinstock (2015)* and our analysis provided data analysis support for this claim.

There were two inflection points and three regression lines in the distribution of TIFtS. The second inflection point was at 93 months (95% CI [87.39–99.01] months), it separated the second and the third power law distributions. Although the second and third distributions were mainly patients with non-synchronous MPMs, our analysis showed a significant enrichment of patients that developed subsequent synchronous MPMs in the second power law distribution than the third distribution (9.1% vs 7.3%, $P < 0.0001$). Furthermore, patients in the third power law distribution were significantly younger (mean initially diagnostic age: 55.01) than patients in the second (mean initially diagnostic

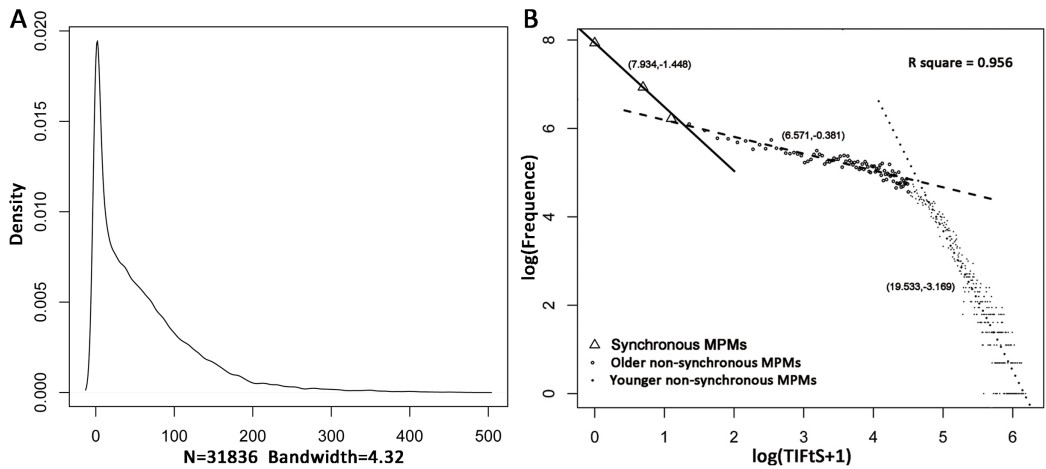

**Figure 1 Distribution of TIFtS.** Kernal density estimation of the distribution of TIFtS. (A) Piece-wise linear regression analysis for the double log transformed distribution of TIFtS. (B) The solid line, dashed line, and dotted line are three regression lines that represent synchronous MPMs, older non-synchronous MPMs, and younger non-synchronous MPMs, respectively. The numbers in the brackets are intercepts and slopes of the regression lines. MPM, multiple primary cutaneous melanoma; TIFtS, time interval between the first and the second primary cutaneous melanomas.

age: 61.01) and the first (mean initially diagnostic age: 60.08) distributions. Thus, the second and third distributions were termed as "older non-synchronous MPMs" and "younger non-synchronous MPMs," respectively (Fig. 1B).

Actually, the indicator variable of the three power law distributions (1, 2, 3 for the first, second, and third distributions, respectively) was time dependent, because it incorporated at least the information of the second tumor, which would happen in the future. To compare the survival of synchronous and non-synchronous MPMs, we first modeled the occurrence of subsequent CM as a time dependent variable and pre-processed the survival data into start–stop format by the following criterion. If time intervals between a tumor and its all neighboring tumors are greater than 2 months, the tumor is defined as non-synchronous MPM, else, that is, at least one neighboring tumor is within 2 months, the tumor is defined as synchronous MPM (Fig. 2). Finally, a patient was divided into several patients according to successive occurrences of synchronous and non-synchronous MPMs (Fig. 2). Because the analysis not just need the first and the second primary CMs but also need all subsequent primary CMs. We further filtered out patients that do not have complete information on subsequent primary CMs. This filtering resulted in 27,877 patients and 57,666 tumors for survival analysis. Of these patients, 10,523 were female and 17,354 were male, the average diagnostic age of the first CM was 59.88 years. At the last follow-up, 20,830 patients were alive and 7,040 were deceased, in which 2,215 deaths were caused by CM.

Univariate Cox proportional hazards model revealed that the occurrence of synchronous MPM was a risk factor for both overall survival (HR = 1.808 (95% CI [1.698–1.925]), $P < 0.0001$) and CM-specific survival (HR = 1.730 (95% CI [1.553–1.928]), $P < 0.0001$). By also modeling age of diagnosis and year of diagnosis as time dependent

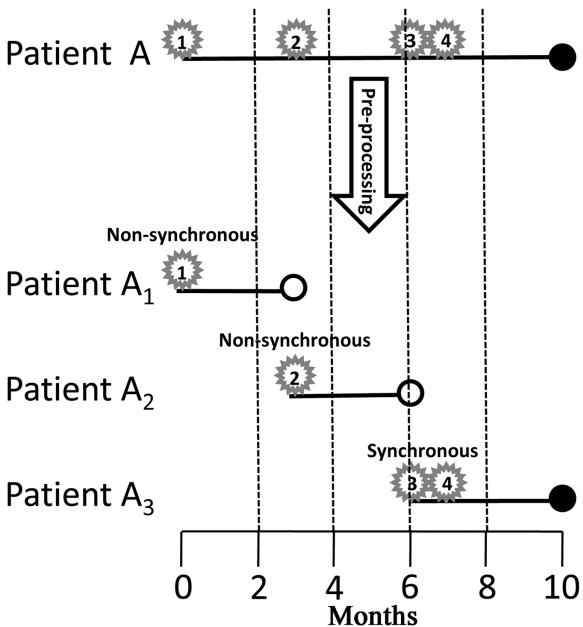

**Figure 2 Pre-processing of the survival data.** Star, number in the star, solid dot, and hollow dot represent tumor, tumor sequence number, death, and censored, respectively. Tumor 1 and 2 are non-synchronous MPMs, and tumor 3 and 4 are synchronous MPMs. Patient A is divided into three patients, the first one starts from the occurrence of tumor 1 and ends up at the occurrence of tumor 2; the second one starts from the occurrence of tumor 2 and ends up at the occurrence of tumor 3; the last one starts from the occurrence of tumor 3 and ends up until death. MPM, multiple primary cutaneous mela-noma.

covariates, multivariate Cox proportional hazards model clustered by patients showed that occurrence of synchronous MPM, older age, latter diagnosis, and male were risk factors for overall survival and CM-specific survival (Table 1). Furthermore, the non-linear dose-response relationship of age at diagnosis and year of diagnosis was explored by a restricted cubic spline analysis with four knots that implemented in the R package "rms." The results showed that both age at diagnosis ($P < 0.001$) and year of diagnosis ($P < 0.0001$) have non-linear associations between overall survival (Fig. 3).

However, the HR of the occurrence of synchronous MPM for overall survival was 2.371 (95% CI [2.108–2.371]) after adjusting for age, and it was 2.213 (95% CI [2.087–2.3461]) after adjusting for age, year, and gender. Thus, age was the main confounding factor for predicting the survival of CM patients, because it leaded to a bigger change to the HR of MPM synchrony compared to year and gender. Finally, Kaplan–Meier analysis revealed that synchronous MPMs showed a significantly inferior overall survival than non-synchronous MPMs after adjusting for age of diagnosis, year of diagnosis, and gender (Fig. 4).

## DISCUSSION

In this study, we analyzed the distribution of TIFtS and found that the distribution could be divided into three power-law distributions. We further define the first power-law distribution as synchronous MPMs, and its inflection point was at 2 months. This cut

**Table 1  Multivariate Cox proportional hazard model clustered by patients.**

|  | HR[OS] | 95% CI[OS] | P[OS] | HR[CMSS] | 95% CI[CMSS] | P[CMSS] |
|---|---|---|---|---|---|---|
| Synchronous MPM | 2.213 | 2.087–2.346 | <0.0001 | 1.980 | 1.776–2.207 | <0.0001 |
| Age at diagnosis | 1.088 | 1.086–1.091 | <0.0001 | 1.054 | 1.050–1.059 | <0.0001 |
| Year of diagnosis | 0.993 | 0.990–0.996 | <0.0001 | 0.987 | 0.979–0.995 | 0.001 |
| Sex | 1.341 | 1.277–1.408 | <0.0001 | 1.427 | 1.301–1.566 | <0.0001 |

**Note:**
HR, hazard ratio; CI, confidence interval; OS, overall survival; CMSS, cutaneous melanoma specific survival; MPM, multiple primary cutaneous melanoma.

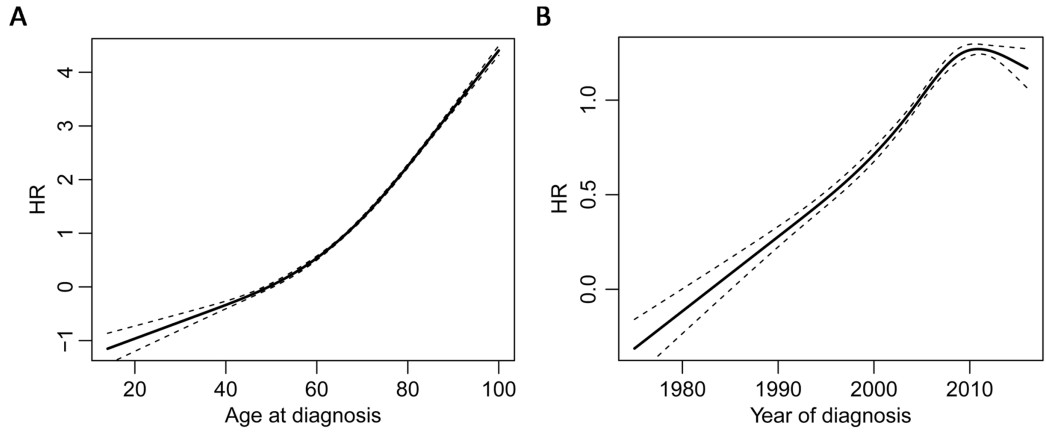

**Figure 3  Non-linear dose-response relationships.** Restricted cubic spline analysis of the association between overall survival and age of diagnosis (A) and the association between overall survival and year of diagnosis (B) the middle solid line indicates the point estimates of hazard ratios and the broken lines indicate the lower and upper limits of the corresponding 95% confidence intervals. Four knots were used for the analysis.

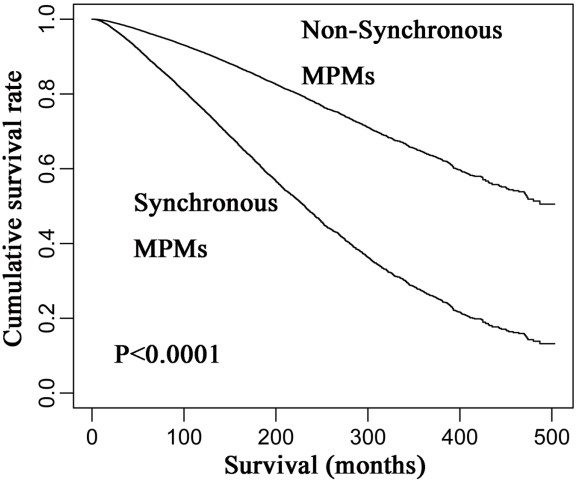

**Figure 4  Kaplan–Meier curves of synchronous and non-synchronous MPMs.** Synchronous MPMs showed a significantly inferior overall survival than non-synchronous MPMs after adjusting for age of diagnosis, year of diagnosis, and gender. MPM, multiple primary cutaneous melanoma.

point was consistent with previous experience (*Pomerantz, Huang & Weinstock, 2015*), and our analysis provided data analysis support to use 2 months to distinguish synchronous MPMs and non-synchronous MPMs. Furthermore, survival analyses revealed that synchronous MPM was a risk factor for CM patient prognosis.

There are two ways to deal with time dependent variables to accommodate the Cox proportional hazard model. A simple way is to define a landmark time to divide patients into two groups. In this approach, patients who receive the intervention prior to the landmark go into the intervention group and those who did not are placed in the comparison group regardless of what happens in the future (*Dirk, 2016*). Indeed, *Grossman et al.'s (2018)* single matching method belongs to this kind. Their landmark time is the TIFtS for each MPM patient and it varies for each patient. However, this landmark method discarded most of the patients from the analysis. The other way is to model the variable as a time dependent variable directly (*Dirk, 2016*). This method avoids discarding any patients and it can include all course of disease. Thus, it is better than the landmark method.

For the Cox proportional hazard model, an important assumption is the proportional hazard. In our analysis, the cumulative incidence plot for synchronous MPM was not parallel (data not shown). This revealed that the proportional hazard assumption was not satisfied. However, the cumulative incidence plot was not crossed and this indicated that although the estimated hazard ratio may be varied with time, the synchronous MPM was still a risk factor for CM patient prognosis.

In addition, pathological variables such as breslow depth, ulceration, mitosis rate, and pathological stage were not analyzed due to too many missing values (*Grossman et al., 2018*) and inaccuracies (*Mayer, Fathi & Norris, 2017*). Thus, the potential pathology of synchronous MPM needs to be illustrated in the future. Furthermore, many molecular events such as mutation (*Demunter et al., 2001*; *Griewank et al., 2014*), copy number variation (*Rákosy et al., 2010*; *Gerami et al., 2011*), epigenetic variation (*Roh et al., 2016*; *Wouters et al., 2017*), expression of genes (*Brown et al., 2012*; *Schramm et al., 2012*) and non-coding RNAs (*Xiong, Bing & Guo, 2019*; *Yang, Xu & Zeng, 2018*) were reported to be involved in the prognosis of CM. Further laboratory studies aimed to investigate the potential molecular mechanisms of synchronous MPM occurrence and its prognostic roles are also in need.

## CONCLUSIONS

In conclusion, this study provided data analysis evidences to distinguish synchronous and non-synchronous MPMs. Although the occurrence of synchronous MPM was a risk factor for CM prognosis, the potential pathological and molecular mechanisms should be illustrated in the future.

## ACKNOWLEDGEMENTS

The authors thank the SEER 18 program made the data of CM available, and all data obtained from SEER keep to the rules for usage and publication of SEER. We also thank the

anonymous reviewers for helpful comments and suggestions. In addition, helpful discussion with Dr. Yuxiang Yao from Lanzhou University is appreciated.

### Funding

This work is supported by the National Natural Science Foundation of China (61772089, 61873221, 61672447), Natural Science Foundation of Hunan Province (2019JJ40325, 2018JJ3566, 2018JJ3565, 2018JJ4058, 2017JJ5036), Scientific Research Fund of Hunan Provincial Education Department (19A043, 19C0177, 19C0181), and Research Project of Hunan Provincial Health Commission (20201952, 20201903).

### Grant Disclosures

The following grant information was disclosed by the authors:
National Natural Science Foundation of China: 61772089, 61873221 and 61672447.
Natural Science Foundation of Hunan Province: 2019JJ40325, 2018JJ3566, 2018JJ3565, 2018JJ4058 and 2017JJ5036.
Scientific Research Fund of Hunan Provincial Education Department: 19A043, 19C0177 and 19C0181.
Research Project of Hunan Provincial Health Commission: 20201952 and 20201903.

### Competing Interests

The authors declare that they have no competing interests.

### Author Contributions

- Jie Xiong conceived and designed the experiments, analyzed the data, prepared figures and/or tables, authored or reviewed drafts of the paper, and approved the final draft.
- Yanlin Su analyzed the data, prepared figures and/or tables, and approved the final draft.
- Zhitong Bing analyzed the data, prepared figures and/or tables, and approved the final draft.
- Bihai Zhao conceived and designed the experiments, authored or reviewed drafts of the paper, and approved the final draft.

### Human Ethics

The following information was supplied relating to ethical approvals (i.e., approving body and any reference numbers):

Our study was exempt from institutional review board oversight, because the SEER 18 database is accessible to the public and the patients in the database are de-identified.

### Data Availability

The raw data can be found at SEER using search terms "CM," "microscopically confirmed," "in situ," and "malignant."

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
