# Peer review of "Survival between synchronous and non-synchronous multiple primary cutaneous melanomas—a SEER database analysis"

_PeerJ, doi:10.7717/peerj.8316_

## Round 0.1 · original submission · Major Revisions

Please make sure that you exhaustively address all the issues raised by the referees including those contained in the annotated manuscript provided by referee #1.

Reviewer 1 ·

Basic reporting

The manuscript needs a linguistic revision

Experimental design

No comment

Validity of the findings

No comment

Additional comments

The distinction among MPM patients between synchronous and non-synchronous disease onset is definitely a valuable scientific target above all when such a distinction is used to assess and test a differential in survival rates. In other words, the detection of synchrony in MPM could have significant implications in clinical practice. Although the Authors used a reliable database (1975-2016 SEER 18 program), applied reasonable inclusion/exclusion criteria (about 32,000 with at least a 5-year follow-up time after diagnosis) and performed appropriate statistical analyses, some remarks have been highlighted to improve the paper. In particular, extensions, modifications and better explanations of statistical methods are suggested. In this way, the information conveyed by this study can reach a larger scientific audience, namely not only researchers with an advanced statistical proficiency but also biomedical investigators and clinical professionals.

Annotated reviews are not available for download in order to protect the identity of reviewers who chose to remain anonymous.

Reviewer 2 ·

Basic reporting

In this manuscript, Authors focused their attention in the development of a criterion to distinguish synchronous and non-synchronous multiple primary cutaneous melanomas (MPCM) and, based on the result obtained, performed survivals analyses between synchronous and non-synchronous MPCM.

The article includes sufficient background related to the unclear and arbitrary time interval considered to define synchronous and non-synchronous MPCM, however literatures related to the different survival of these patients in relation to their multiple primary CM should be added.
Figure 1: Authors stated: “The numbers in the brackets are intercepts and slopes of the regression lines”. It is not so clear to me the intercept of the solid line. The number is 7.9 but the intercept should be more than 8.
Figure 2: X axis labeled is missing.
Fig 3: Results about the inferior overall survival should be added here and in the text.

Experimental design

The Research question raised by the authors is well defined and the size of investigated features is considerable to address the scope of the study. However, overall the methods utilized by the authors should be more clearly defined. For example (and this is just an example) the algorithm utilized to optimize the cut points of the distribution is not specified.

Validity of the findings

The high number of patients analyzed contributes to give value to the obtained results; however, as correctly pointed out by the authors, the investigation of molecular (i.e., genetic and epigenetic) characteristics of the different lesions could add important value to the findings obtained. In addition, information on the possible adjuvant therapy made by patients who developed a subsequent primary CM, as well as information about the pathological staging of the primary lesion are not mentioned. These additional clinical data could indeed affect patients' survival.

Additional comments

The authors have well addressed the question raised about the time interval required to define a second primary tumor as synchronous or non-synchronous with the first one. However, no pathologic, genetic or epigenetic information about these lesions are described. Although the authors pointed out these limitations in the discussion, such data would have improved the novelty of the work.

Reviewer 3 ·

Basic reporting

The manuscript needs some minor English revision.

Experimental design

No comment.

Validity of the findings

No comment.

---

## Round 0.2 · accepted · Accept

You have adequately addressed the issues raised by the referees.

Reviewer 1 ·

Basic reporting

no comment

Experimental design

no comment

Validity of the findings

no comment

Additional comments

no comment

Reviewer 2 ·

Basic reporting

In this manuscript, Authors focused their attention in the development of a criterion to distinguish synchronous and non-synchronous multiple primary cutaneous melanomas (MPCM) and, based on the result obtained, performed survivals analyses between synchronous and non-synchronous MPCM.

Experimental design

No comment

Validity of the findings

No comment

Additional comments

Authors did a good job of incorporating important review elements.